# Direct and Indirect Effects of Youth Sports Participation on Emotional Intelligence, Self-Esteem, and Life Satisfaction

**DOI:** 10.3390/sports12060155

**Published:** 2024-06-03

**Authors:** Helder Miguel Fernandes, Henrique Costa, Pedro Esteves, Aristides M. Machado-Rodrigues, Teresa Fonseca

**Affiliations:** 1Polytechnic Institute of Guarda, 6300-559 Guarda, Portugal; henriquefcosta@gmail.com (H.C.); ptesteves@ipg.pt (P.E.); tfonseca@ipg.pt (T.F.); 2Sport Physical Activity and Health Research & INnovation CenTer (SPRINT), 6300-559 Guarda, Portugal; 3Faculty of Sport Sciences and Physical Education, University of Coimbra, 3040-248 Coimbra, Portugal; a.machado-rodrigues@fcdef.uc.pt; 4Research Centre of Sport and Physical Activity (CIDAF-UC), University of Coimbra, 3040-248 Coimbra, Portugal

**Keywords:** adolescents, sports participation, direct and indirect effects, well-being, mental health

## Abstract

The present study investigated the mediating effects of emotional intelligence and self-esteem between youth sports participation and life satisfaction, as well as the comparative effects of different types of sports involvement (team, individual, and non-participation) on these selected variables. A sample of 1053 Portuguese adolescents (612 girls and 441 boys), aged between 12 and 18 years (*M* = 14.40; *SD* = 1.55), completed the following self-report measures: the Wong and Law Emotional Intelligence Scale, the Rosenberg Self-Esteem Scale, and the Satisfaction with Life Scale. The structural equation modeling results indicated a complete mediating role of two emotional intelligence dimensions (use of emotion and self-emotional appraisal) and self-esteem in the relationship between sports participation and adolescents’ life satisfaction. Team sport participants reported higher emotional intelligence and self-esteem scores than their non-sport participant counterparts who revealed lower levels of emotion use than their individual sport participant peers. These findings provide novel insights into the potential emotional and psychological mechanisms underlying the association between youth sports participation and life satisfaction.

## 1. Introduction

Regular and sufficient physical activity (PA) during adolescence has been consistently associated with important physical, cognitive, psychological, and social benefits [1,2]. Supported by this evidence, several international organizations have published guidelines that recommend that school-aged youth should accumulate at least 60 min of moderate- to vigorous-intensity physical activity (MVPA) every day for health promotion purposes [3,4]. However, several epidemiological studies have shown that only a minority of young people achieve sufficient PA levels [5,6,7], with Portuguese adolescents showing a similar trend [8,9]. Thus, participation in organized youth sports has been suggested as an important dimension of PA which may substantially contribute to higher MVPA levels in adolescents [10], with the potential to promote specific and additional physical, psychological, and social benefits over other contexts of PA [11].

Participation in youth sports is a type of leisure-time PA that usually occurs in organized settings under the supervision of an adult or youth leader (i.e., coach) and involves rules and formal practice in training and competition sessions, either individually or in small groups or teams [10,12]. In recent decades, sports participation has been considered an important part of adolescents’ lives, encompassing important experiences and effects in terms of health and developmental outcomes [13]. Engaging and participating in this structured environment may contribute to the personal and social development of young people by providing opportunities to experience both positive and negative situations. It is necessary, therefore, to better understand the potential emotional and psychological mechanisms underlying the relationship between sports participation and well-being during adolescence [10,14].

Subjective well-being refers to how individuals experience and evaluate their lives in terms of affective and cognitive aspects [15]. The affective dimension includes both positive (e.g., joy, optimism) and negative (e.g., anger, sadness) individual emotional responses, whereas life satisfaction encompasses the cognitive, judgmental component of subjective well-being [16]. Unlike emotional responses, satisfaction with life is considered a more stable indicator of adolescents’ psychological well-being and is positively associated with a broad spectrum of intrapersonal and interpersonal beneficial outcomes [17]. Moreover, meta-analytic findings showed no sex differences in life satisfaction among youths [18]. However, cross-national and nationally representative longitudinal evidence from 43 countries participating in the Health Behavior in School-aged Children (HBSC) study demonstrated that life satisfaction declined between the ages of 10 and 16 years in all examined countries [19]. This phenomenon can be driven by economic, mental, and social factors such as income quality, mental disorders, or gender equality [20]. Sports participation has been identified as a relevant context for well-being and mental health promotion during adolescence [21], with several studies showing a positive and direct effect of youth sports involvement on satisfaction with life [16,22,23]. Interestingly, there is evidence suggesting that this association depends on the effect of mediating influences [24]. Since sports participation has the potential to enhance adolescents’ life satisfaction and well-being, it is crucial to better examine the potential influences and mechanisms involved in the aforementioned relationship.

Among the possible emotional and psychological explanatory mechanisms of PA on mental health in young people, self-knowledge, emotional management, self-perception, and self-esteem stand out as derivates of regular sports participation [10,25]. Organized sports require individuals to successfully cope with emotional and psychological difficulties such as pressure, stress, burnout, and immoral behaviors [12], which include perceiving, understanding, and regulating their emotions and those of others, namely, teammates, coaches, opponents, referees, and even spectators. According to the four-branch model of Mayer and Salovey [26], emotional intelligence is a set of interrelated emotion-related abilities to (i) perceive emotions in oneself and others accurately, (ii) use emotions to facilitate thinking and decision-making, (iii) understand emotions, emotional language, and the signals conveyed by emotions, and (iv) manage emotions to attain goals. Substantial research has demonstrated that regular and continued sports participation may help young athletes become more proficient in understanding and regulating their emotions, as well as in managing others’ emotions [27,28,29]. This successful emotional regulation and management may contribute to better interaction, cooperation, and support in group (team) members [30], which in turn may improve adolescents’ self-image and self-esteem [31]. Self-esteem consists of a global assessment of the perceived self, involving one’s positive and negative evaluations and attitudes towards oneself (i.e., self-worth and self-acceptance) [32,33]. Several studies have demonstrated that self-esteem is a protective factor for mental health, serving as a major determinant of well-being during adolescence (e.g., [34,35]). In addition to the consensus regarding the beneficial effects of PA and sports participation on self-esteem [11,36,37], studies have shown that emotional intelligence also plays a significant antecedent role in self-esteem development [38,39]. In particular, the dimensions of attention, clarity, and repair of emotions are positively associated with adolescents’ self-perceptions and self-worth. Moreover, there is also consistent evidence of the mediating effects of self-esteem on the association between emotional intelligence dimensions and life satisfaction in young people [40,41], suggesting that adolescents with greater emotional intelligence can maintain greater self-worth by better dealing with situational and emotional demands [38], which in turn leads to increased life satisfaction and overall well-being [40,42].

Another source of variation that has received less attention is the effects of different types of youth sports involvement on selected outcomes, such as emotional and mental health. A considerable number of studies have shown that team sport participation promotes better emotional and mental health outcomes than individual or non-sport participation [11,21,22,43,44], whereas other studies have indicated no differences by type of sport [12,27,45]. Therefore, the current study also sought to clarify previously mixed findings regarding the effects of different types of youth sports involvement on selected outcomes [14].

Although the literature is clear regarding the associations between sports participation, emotional intelligence, self-esteem, and life satisfaction, far less is known about the processes and mechanisms underlying these sequential relationships. Therefore, the main purpose of the present study was to investigate the mediating effects of emotional intelligence and self-esteem on the relationship between youth sports participation and life satisfaction through structural equation modeling (SEM). Figure 1 presents the hypothesized conceptual model of the present study.

Considering previous studies that have demonstrated causal chain mediating effects of emotional intelligence and self-esteem [29,46] or serial links between emotional intelligence, self-esteem, and life satisfaction [38,40,41], it was hypothesized in the present study that sports participation might have a positive, direct effect on adolescents’ life satisfaction, as well as positive, indirect effects through serial mediating effects of the emotional intelligence dimensions and global self-esteem. Moreover, this study also has a secondary aim of examining the effects of being involved in different types of sports (individual or team) on emotional intelligence, self-esteem, and life satisfaction levels.

## 2. Materials and Methods

The current study is a cross-sectional survey based on a quantitative approach conducted among adolescents enrolled in public schools in the central region of Portugal.

### 2.1. Participants

A total of 1074 adolescents were included in this study. After excluding participants with missing data, the final sample included 1053 adolescents (612 girls and 441 boys) aged between 12 and 18 years (*M* = 14.40; *SD* = 1.55). The participants were 7th- to 12th-grade students attending different Portuguese public schools. Nearly one-third of the adolescents (*n* = 354, 33.6%) reported participating in organized sports, while non-participants represented the greatest proportion of the sample (*n* = 699, 66.4%). With regard to sport type, 197 adolescents (18.7%) reported participating in team sports, whereas 157 (14.9%) participated in individual sports. The inclusion criteria in the study were as follows: (i) male or female participants aged between 12 and 18 years, (ii) school attendance in one of the selected classes, and (iii) signed informed consent by the parents and/or the legal guardian. Moreover, a large sample size (*n* > 1000) was planned a priori, as recommended in the literature for more complex models [47] using SEM.

### 2.2. Instruments

Initially, participants provided self-report information regarding their sex, age, and sports engagement. Participation in organized sports outside of school was assessed by a single-item question, with a binary no/yes response. Organized sport was defined as sports activities guided by coaches and practiced in sports clubs. Responses to the type of sports participation question were classified as individual (e.g., gymnastics, swimming, track and field, martial arts) or team (e.g., soccer, basketball, futsal, handball, volleyball).

Next, participants completed a battery of self-report questionnaires measuring emotional intelligence, self-esteem, and life satisfaction.

#### 2.2.1. Emotional Intelligence

A validated Portuguese version [48] of the Wong and Law Emotional Intelligence Scale [49] was used to measure the perceived levels of emotional intelligence. This multidimensional instrument includes 16 items grouped into four dimensions: (i) self-emotional appraisal (SEA, 4 items), (ii) others’ emotional appraisal (OEA, 4 items), (iii) use of emotion (UOE, 4 items), and (iv) regulation of emotion (ROE, 4 items). The items were rated on a 5-point Likert scale ranging from 1 (strongly disagree) to 5 (strongly agree). The total score of the four dimensions was computed as the sum of the scores of the items included. Possible scores range from 5 to 20, with higher scores indicating higher levels of emotional intelligence. The psychometric properties of the WLEIS for the present study were good. The four-factor oblique model, with correlated errors between items 9 and 12 (both included in the use of emotion dimension), revealed an adequate fit, χ^2^(97) = 418.82, *p* < 0.001, CFI = 0.940, RMSEA = 0.056 (90% CI = 0.051−0.062), SRMR = 0.059. Standardized factor loadings ranged between 0.42 and 0.86. Omega reliability values showed adequate internal consistency for all dimensions, namely, self-emotional appraisal (ω = 0.75), others’ emotional appraisal (ω = 0.75), use of emotion (ω = 0.79), and regulation of emotion (ω = 0.81).

#### 2.2.2. Self-Esteem

A validated Portuguese version [33] of the Rosenberg Self-Esteem Scale [32] was administered to assess adolescents’ self-esteem levels. This scale consists of 10 items, which were rated on a 4-point Likert scale ranging from 1 (strongly disagree) to 4 (strongly agree). Five of the items are positively worded, while the remaining five are negatively worded. The total score of the scale was computed through the sum of the scores of all items after reverse-coding the negative items. Possible scores range from 10 to 40, with higher scores indicating higher levels of perceived self-esteem. In the present study, the one-factor model, with correlated errors among positively worded items, showed a better fit, χ^2^(25) = 59.38, *p* < 0.001, CFI = 0.989, RMSEA = 0.036 (90% CI = 0.024–0.048), SRMR = 0.021. Standardized factor loadings ranged between 0.42 and 0.78. The model-based omega reliability was acceptable (ω = 0.82).

#### 2.2.3. Life Satisfaction

A validated Portuguese version [50] of the Satisfaction with Life Scale [15] was used to measure adolescents’ overall life satisfaction. This unidimensional scale includes 5 items rated on a 7-point Likert scale ranging from 1 (strongly disagree) to 7 (strongly agree). The item scores were summed to yield a composite score, with possible scores between 5 and 35. Higher scores indicate better life satisfaction. In the present study, the one-factor model, with correlated errors between items 2 and 5, showed a good fit, χ^2^(4) = 9.77, *p* = 0.045, CFI = 0.997, RMSEA = 0.037 (90% CI = 0.005–0.067), SRMR = 0.012. Standardized factor loadings ranged from 0.60 to 0.83. Internal consistency analysis revealed that the omega reliability was high (ω = 0.84).

### 2.3. Procedures

The study sample was selected using multistage sampling procedures. Initially, public schools were purposively chosen from selected districts located in the Portuguese midlands. After receiving the school principals’ approval to collect the data, one to two classes from the 7th to 12th grades (corresponding to the 3rd cycle of elementary school or high school) were randomly selected from each school by lottery method.

Before the data were collected, written consent was obtained from the adolescents’ parents/guardians. Subsequently, participants were verbally informed about the procedure, as well as the instructions to complete the questionnaires. Participation in the survey was voluntary and the collected data were anonymous and confidential. Questionnaires were administered individually in quiet classroom conditions.

This study conformed to the Declaration of Helsinki and was approved by the ethics committee of the Portuguese Directorate-General for Education (Study Registration N° 0395700001/MIME).

### 2.4. Statistical Analyses

A preliminary inspection of the dataset was performed to screen the data for accuracy and missing values.

Descriptive statistics such as the means (*M*), standard deviations (*SD*), 95% confidence intervals (CI), and frequency and percentage (%) were calculated, as well as bivariate correlations (point biserial and Pearson) between variables under analysis. Univariate skewness and kurtosis estimates were computed for all items to determine the (non)normality of the data. The values ranged between −1.76 and 3.47, which suggests a small departure from normality [47,51]. These analyses were conducted using IBM SPSS Statistics for Windows, version 23 (IBM Corp., Armonk, NY, USA). The level of statistical significance was set at 5%.

Next, SEM was employed to test the direct and indirect relationships of the hypothesized model, which was examined using a two-step approach [52]. First, confirmatory factor analyses (CFA) were performed separately for each scale and then for the full measurement model. The internal consistency of the latent variables was analyzed by computing McDonald’s omega coefficient (ω), with values of 0.70 or higher considered satisfactory. Since sports participation was measured using a dichotomous response format (0 = no; 1 = yes), this variable was included only in the next (structural) step. After obtaining an adequate fit of the measurement model, the second step of the model building/testing approach included assessing the fit of the structural model to the data. Both absolute and incremental fit indices were used to estimate the adequacy of the measurement and the structural models [47], namely, χ^2^, with its degrees of freedom and significance (*p*) value; the root mean square error of approximation (RMSEA), with its 90% CI; the comparative fit index (CFI); and the standardized root mean square residual (SRMR). The following cutoff values were adopted to indicate an acceptable fit of the model to the data [40,46]: χ^2^/df < 3, RMSEA (CI90%) < 0.06 (<0.10), CFI ≥ 0.90, and SRMR < 0.08. An inspection of the normalized estimate of Mardia’s coefficient showed a very large result (226.44, critical ratio = 81.23). To address this significant departure from multivariate normality, all SEM analyses were performed employing the maximum likelihood bootstrapping procedure using 5000 bootstrap replication samples and a 95% bias-corrected CI. The same procedure was used to analyze the direct and indirect effects of sports participation on emotional intelligence, self-esteem, and life satisfaction by computing estimates, standard errors, and confidence intervals (95% CIs) for all relevant effects. The (in)direct effects were considered significant when zero was not included in the 95% CI [47,51]. These analyses were conducted using IBM SPSS Amos for Windows, version 23 (IBM Corp., Armonk, NY, USA). The level of statistical significance was set at 5% or at a 95% CI.

Finally, one-way ANOVAs were also conducted to further examine the effects of sports type (which were arranged into three groups: non-sport participation, individual participation, and team participation) on the dependent variables. Due to the unequal sample sizes of the three groups, the Games–Howell post hoc test was used for multiple comparisons. Estimates of effect size (partial eta squared: η_p_^2^) were used to interpret the magnitude of the differences between groups as follows: small (η_p_^2^ > 0.01), medium (η_p_^2^ > 0.06), or large (η_p_^2^ > 0.14) [53]. These analyses were conducted using IBM SPSS Statistics for Windows, version 23 (IBM Corp., Armonk, NY, USA). The level of statistical significance was set at *p* ≤ 0.05.

## 3. Results

### 3.1. Descriptive and Correlational Analyses

Table 1 presents the descriptive statistics and intercorrelations for all the variables used in the study.

On average, participants reported moderate levels of perceived emotional intelligence (self-emotional appraisal, others’ emotion appraisal, and use of emotion), self-esteem, and life satisfaction. Lower mean scores were found for the regulation of emotion scale of emotional intelligence (although above the midpoint).

Point biserial correlations revealed small, significant relationships between sports participation and self-emotional appraisal, use of emotion, self-esteem, and life satisfaction dimensions. Pearson correlations indicated moderate associations between the emotional intelligence scales, with the exception of others’ emotional appraisal (which was correlated only with the use of the emotion scale). Pearson’s correlation analysis also revealed moderate to high correlations between self-esteem and life satisfaction scores, as well as between these two variables and the three emotional intelligence scales (self-emotional appraisal, use of emotion, and regulation of emotion).

### 3.2. Measurement Model

Prior to testing the initially hypothesized model, the factorial structure of each model’s components was analyzed. The results (presented in the Section 2.2) showed that all the scales fit the data well (CFI ≥ 0.94 and RMSEA < 0.06) and that all the latent factors displayed adequate composite reliability (ω ≥ 0.70).

Next, the full measurement model was tested via CFA, which included covariances between all the latent variables. The results showed that the model provided a satisfactory fit to the data (χ^2^(407) = 1331.61, *p* < 0.001; CFI = 0.921; RMSEA = 0.046 (90% CI = 0.044−0.049); SRMR = 0.061). All factor loadings and correlations between factors or error terms were significant, except for relationships between others’ emotion appraisal, regulation of emotion (*p* = 0.316), and self-esteem (*p* = 0.190). Therefore, these two covariances were dropped from the measurement model. The results showed that this slightly modified version of the measurement model retained an acceptable fit to the data (χ^2^(409) = 1333.70, *p* < 0.001; CFI = 0.921; RMSEA = 0.046 (90% CI = 0.044−0.049); SRMR = 0.062).

### 3.3. Structural Model

The second stage of the model-building/testing approach involved the inclusion of the observed variable of sports participation as an exogenous variable in the hypothesized path model. Based on the aforementioned results, the disturbance terms between the emotional intelligence scales were allowed to correlate, except for the other’s emotion appraisal and the regulation of emotion. Although the path model fit the data adequately, there was room for improvement: χ^2^(433) = 1383.07, *p* < 0.001; CFI = 0.919; RMSEA = 0.046 (90% CI = 0.043−0.048); SRMR = 0.060. The hypothesized direct paths from sports participation to others’ emotion appraisal, regulation of emotion, self-esteem, and life satisfaction dimensions were not significant and were therefore dropped. The direct paths between others’ emotional appraisal, self-esteem, and life satisfaction were also not significant and were removed from the model. In addition, the nonsignificant direct path from the regulation of emotion to life satisfaction was also dropped. Consequently, it was possible to determine that the others’ emotional appraisal latent variable was not significantly related to any other variables (except for the disturbance terms with the other emotional intelligence dimensions), and for parsimony purposes, it was subsequently removed from the final model. The new results showed that the fit of the revised model to the data improved: χ^2^(328) = 1015.65, *p* < 0.001; CFI = 0.935; RMSEA = 0.045 (90% CI = 0.042–0.048); SRMR = 0.058.

The standardized solution for the final direct and indirect effects model is presented in Figure 2. For clarity purposes, parameters for the measurement portion of the model and error/disturbance terms are not presented.

Sports participation was positively associated with self-emotional appraisal and the use of emotion. All three emotional intelligence factors included in the model were positively associated with self-esteem, whereas only self-emotional appraisal and the use of emotion variables had direct, positive effects on life satisfaction. As expected, self-esteem was positively related to life satisfaction.

### 3.4. Indirect Effects

The results of the unstandardized indirect effects and the lower and upper limits of bootstrap-generated bias-corrected 95% CIs of the mediated effects for the final structural model are presented in Table 2.

The unstandardized indirect effects revealed that sports participation had positive, indirect effects on self-esteem, which were mediated by the self-emotional appraisal and use of emotion (emotional intelligence) dimensions. Moreover, the results also showed that sports participation had a significant indirect effect on life satisfaction, which was realized through serial mediation of emotional intelligence dimensions (self-emotional appraisal and use of emotion) and self-esteem. On the contrary, there were no significant effects of sports participation on life satisfaction when emotional intelligence and self-esteem were considered mediators. Notably, the unstandardized estimates of the indirect effects showed that the use of the emotion pathway had a greater contribution to self-esteem (87.5%) and life satisfaction (78.9%) outcomes when influenced by sports participation.

### 3.5. Effects of Sport Type on Emotional Intelligence, Self-Esteem, and Life Satisfaction

The influence of sports participation on adolescents’ psychological development was further investigated in this study by examining the effects of being involved in distinct types of sports (individual and team) on emotional intelligence, self-esteem, and life satisfaction levels. Previously, the univariate normality assumption was verified for all latent variables, with values of skewness and kurtosis ranging from −1.23 to 1.62.

Table 3 summarizes the descriptive statistics for the dependent variables by type of sports participation.

Separate one-way ANOVAs revealed that the effect of sport type was significant for self-emotional appraisal (*F*(2, 1050) = 5.95, *p* = 0.003, η_p_^2^ = 0.01, power = 0.88; use of emotion, *F*(2, 1050) = 23.50, *p* < 0.001, η_p_^2^ = 0.04, power > 0.99; regulation of emotion, *F*(2, 1050) = 3.59, *p* = 0.028, η_p_^2^ = 0.01, power = 0.67; and self-esteem, *F*(2, 1050) = 11.38, *p* < 0.001, η_p_^2^ = 0.02, power = 0.99). No significant effect was found on others’ emotional appraisal (*F*(2, 1050) = 1.64, *p* = 0.194, η_p_^2^ = 0.00, power = 0.35) or life satisfaction (*F*(2, 1050) = 2.94, *p* = 0.053, η_p_^2^ = 0.01, power = 0.57).

Post hoc tests indicated a trend toward increased self-emotional appraisal scores (*p* < 0.001), use of emotion (*p* < 0.001), regulation of emotion (*p* = 0.017), and self-esteem (*p* < 0.001) in team sport participants compared to non-sport participants. On the other hand, individual sports participants reported higher levels of use of emotion (*p* < 0.01) than non-sport participants. No significant differences were found between the individual and team sport athletes.

## 4. Discussion

The main purpose of the present study was to investigate the direct and indirect (mediating) effects underlying the association between youth sports participation and life satisfaction. More precisely, the emotional intelligence and self-esteem dimensions were hypothesized to mediate this relationship in the aforementioned serial mediation model (Figure 1). The findings of this study revealed that emotional intelligence dimensions (self-emotional appraisal and use of emotion) and self-esteem play chain mediating roles between youth sports participation and life satisfaction, providing support for the hypothesized model. To our knowledge, this is the first study to explore and demonstrate such sequential mediating effects between sports participation, emotional intelligence, self-esteem, and life satisfaction of adolescents.

### 4.1. Effects of Sports Participation on Emotional Intelligence, Self-Esteem, and Life Satisfaction

One of the main findings of the study indicated that the direct effect of sports participation on life satisfaction was nonsignificant, suggesting the existence of a complete mediating effect. More specifically, the small association between youth sports participation and life satisfaction (see correlation coefficient in Table 1) was not supported when two mediators (emotional intelligence and self-esteem) were considered in the path model. This finding is consistent with recent research [24,54] and suggests that adolescents participating in organized sports are more satisfied with their lives when they are (more) able to perceive and use their emotions and, consequently, experience a more positive global evaluation of themselves [55]. Previous research has demonstrated that regular sports participation may foster opportunities for social interactions and relatedness in a formal/structured context under the supervision of coaches [43,56], helping adolescents to better perceive, understand, and use their emotions to facilitate performance, as well as experiencing more feelings of self-acceptance and self-worth [10,25]. While previous research has suggested that PA and sports participation may also be associated with others’ emotional appraisal and regulation of emotion [27,28,29], it was unexpected to find no significant associations or effects in the present study. Possible explanations for this finding may include the fact that interpersonal emotional regulation tends to be perceived by young athletes as less influential for performance outcomes [57] or the fact that the domain of regulation of emotion requires more complex self-regulatory processes and abilities insufficiently developed by adolescents, as shown by the lower mean scores presented in Table 1.

An interesting finding was also the greater contribution of the use of emotion mediating pathway between sports participation, self-esteem, and life satisfaction. More than perceiving and understanding emotions, the results of the present study indicate that participating in organized sports contributes to adolescents developing their ability to use and direct their emotions toward self-directed, constructive activities to facilitate higher achievements and performance, which in turn contributes to their self-esteem and well-being. This finding confirms and extends previous studies that have shown stronger associations between the use of emotion, self-esteem, and life satisfaction in adult athletes [42,55] when compared to the remaining emotional intelligence dimensions. Interestingly, results showed that the indirect effect of use of emotion on life satisfaction via self-esteem was larger than the direct effect (59.7% vs. 40.3%). This suggests that feelings of self-worth and self-acceptance (global self-esteem) play a significant, but complex role in the relationship between emotional intelligence and life satisfaction [38,39]. Nevertheless, more research is needed to investigate the multidimensional and specific effects of different emotional intelligence domains throughout adolescence. Previous research, using SEM, has mainly considered and investigated the (in)direct effects of emotional intelligence as a global composite factor [29,40,58], making it more difficult to compare the current findings with those of others.

### 4.2. Effects of Sport Type on Emotional Intelligence, Self-Esteem, and Life Satisfaction

Additional interesting results of the present study demonstrated that although both team and individual sport participation were associated with higher levels of use of emotion, adolescents involved in team sports also reported better levels of other emotional intelligence dimensions (self-emotional appraisal and regulation of emotion) and self-esteem than their non-participant peers. These findings corroborate previous studies that showed a higher effect of team sport participation on different emotional and mental health outcomes [11,14,21,22,43,44], which may be explained by the social interaction and relatedness aspects of team sports involvement when compared to non-participation in sports. Other studies have suggested that team sport participation has the potential to increase self-esteem and well-being through the enhancement of perceived social acceptance, functioning, and integration [30,31,45,59], providing a unique developmental context for adolescents. Nonetheless, since no significant differences were observed between team sports and individual sports participants in the present study, additional research exploring different types of sports involvement is encouraged, considering other potential moderators (e.g., sex, social support, coach-athlete relationships, indoor/outdoor sports).

### 4.3. Strengths and Limitations

The current study has several strengths and limitations to be acknowledged. This study is the first to explore and identify the emotional and psychological mechanisms underlying the association between youth sports participation and life satisfaction, using emotional intelligence dimensions and self-esteem as mediators. An additional strength was the use of a large sample size (*n* > 1000 adolescents). On the other hand, the study limitations include the cross-sectional design of this study, which prevents causal inferences from being made. Moreover, the unequal size of groups regarding different sports involvement (team, individual, and non-participation) does not allow for the measurement and structural invariance testing of the hypothesized conceptual model. Therefore, future studies should aim to overcome this limitation by recruiting equal-sized study groups in terms of sports involvement.

## 5. Conclusions

The results of the current study indicate that participating in organized youth sports contributes to the development of specific emotional intelligence dimensions (use of emotion and self-emotional appraisal), which in turn leads to increased levels of self-esteem and life satisfaction. These findings demonstrate that sports participation is positively associated with different adolescents’ developmental outcomes (emotional functioning, self-worth, and well-being) through specific direct and indirect effects, providing novel insights into the emotional and psychological explanatory pathways and mechanisms underlying the association between youth sports participation and life satisfaction. Moreover, the results also showed that both team and individual sports participation were associated with higher levels of use of emotion in comparison to non-participants, whereas team sport participants also revealed better levels of self-emotional appraisal, regulation of emotion, and self-esteem.

Sports coaches and researchers are encouraged to implement emotional education programs within youth sports contexts to foster and promote emotional and psychological well-being during adolescence. These interventions should include the assessment of short- and long-term consequences and incorporate some other more established psychological training techniques, such as the training of coping skills and emotion regulation [27].

## Figures and Tables

**Figure 1 sports-12-00155-f001:**
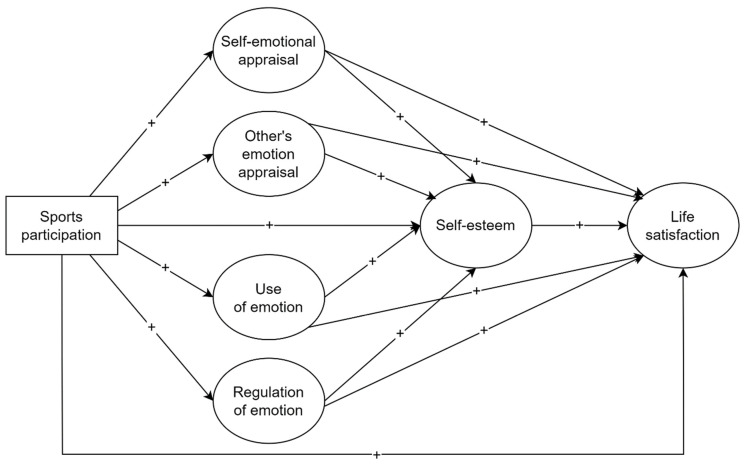
The hypothesized conceptual model.

**Figure 2 sports-12-00155-f002:**
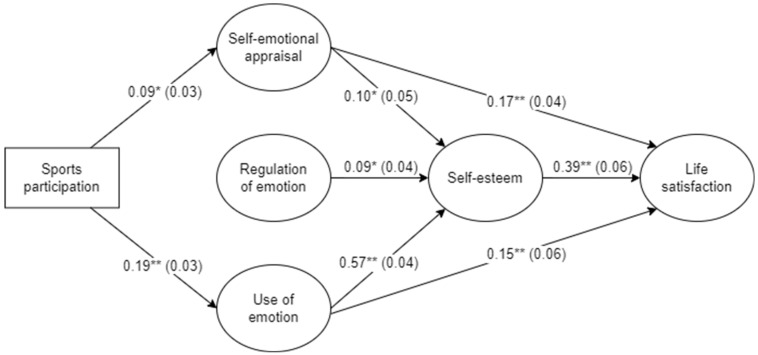
Final direct and indirect effects model. All standardized estimates are significant (* *p* < 0.05, ** *p* < 0.01). The bias-corrected bootstrap estimate of the standard error for each parameter is shown in parentheses. The disturbance terms of the three emotional intelligence factors were allowed to correlate, and the results were as follows: *r* self-emotional appraisal − regulation of emotion = 0.44, *p* < 0.001; *r* self-emotional appraisal − use of emotion = 0.50, *p* < 0.001; and *r* regulation of emotion − use of emotion = 0.42, *p* < 0.001.

**Table 1 sports-12-00155-t001:** Descriptive statistics and intercorrelations of the study variables.

Variable	Range	*M* ± *SD*	Mean 95% CI	1	2	3	4	5	6
1. Sports participation	0–1	---	---	---					
2. EI Self-emotional appraisal	5–20	15.53 ± 2.46	15.39–15.69	0.08 *	---				
3. EI Others’ emotion appraisal	5–20	15.47 ± 2.40	15.35–15.65	−0.03	0.19 **	---			
4. EI Use of emotion	5–20	15.34 ± 2.78	15.18–15.52	0.20 **	0.45 **	0.11 **	---		
5. EI Regulation of emotion	5–20	13.44 ± 3.18	13.27–13.66	0.04	0.37 **	0.01	0.36 **	---	
6. Self-esteem	10–40	30.95 ± 4.60	30.70–31.27	0.14 **	0.42 **	0.06	0.57 **	0.32 **	---
7. Life satisfaction	5–35	27.78 ± 5.44	27.44–28.11	0.06 *	0.33 **	0.09 **	0.41 **	0.29 **	0.49 **

* *p* < 0.05, ** *p* < 0.01; EI = emotional intelligence.

**Table 2 sports-12-00155-t002:** Unstandardized parameter estimates of indirect effects.

Parameter	Estimate (Standard Error)	Bootstrap Bias-Corrected 95% CI (Lower, Upper)	*p*
*Specific indirect effects*			
Sports participation → EI SEA → Self-esteem	0.01 (0.00)	0.00, 0.02	0.025
Sports participation → EI SEA → Life satisfaction	0.03 (0.01)	0.01, 0.07	<0.001
Sports participation → EI SEA → SE → Life satisfaction	0.01 (0.01)	0.00, 0.02	0.022
Sports participation → EI UoE → Self-esteem	0.07 (0.02)	0.05, 0.11	<0.001
Sports participation → EI UoE → Life satisfaction	0.06 (0.03)	0.02, 0.12	0.006
Sports participation → EI UoE → SE → Life satisfaction	0.09 (0.02)	0.05, 0.14	<0.001
*Total indirect effects*			
Sports participation → Self-esteem	0.08 (0.02)	0.05, 0.11	<0.001
Sports participation → Life satisfaction	0.19 (0.04)	0.12, 0.26	<0.001

Note: EI = emotional intelligence; SEA = self-emotional appraisal; UoE = use of emotion; SE = self-esteem.

**Table 3 sports-12-00155-t003:** Descriptive statistics for the dependent variables by type of sports participation.

Variable	Non-Sport Participation *M* ± *SD*	Individual Sports Participation *M* ± *SD*	Team Sports Participation *M* ± *SD*
EI Self-emotional appraisal	15.39 ± 2.50	15.49 ± 2.65	16.06 ± 2.04
EI Others’ emotion appraisal	15.53 ± 2.46	15.58 ± 2.37	15.20 ± 2.19
EI Use of emotion	14.95 ± 2.76	15.73 ± 2.99	16.39 ± 2.32
EI Regulation of emotion	13.35 ± 3.22	13.22 ± 3.44	13.99 ± 2.76
Self-esteem	30.50 ± 4.70	31.34 ± 4.54	32.21 ± 4.02
Life satisfaction	27.53 ± 5.59	27.81 ± 5.71	28.62 ± 5.44

Note: EI = emotional intelligence; SEA = self-emotional appraisal.

## Data Availability

Data is available upon request from the corresponding author due to privacy and ethical restrictions.

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
