# Peer review of "Direct and Indirect Effects of Youth Sports Participation on Emotional Intelligence, Self-Esteem, and Life Satisfaction"

_sports, 2024, doi:10.3390/sports12060155_

Round 1

Reviewer 1 Report

Comments and Suggestions for Authors

Dear Authors,

I am delighted to review an original article draft entitled "Direct and indirect effects of youth sports participation on emotional intelligence, self-esteem and life satisfaction" focused on the mediating effects of emotional intelligence and self- 13 esteem between youth sports participation and their life satisfaction, as well as the comparative 14 effects of different types of sports involvement.

In general, the paper is logically structured and well-written. The abstract is informative. However, there is room for improvement.

I have spotted a few redundant capital letters, for example, in line 56.

Due to complexity, the variable "satisfaction" could be better introduced in the introduction, referring to recent studies and conceptual works. 

(67-107) in general, the proposed theoretical framework could be enhanced by deeper diving into self-concept and self-esteem, e.g. please see Glebova&LopezCarril,2023  and Hidayat&Yudiana&Hambali&Sultoni el al.,2023

Self-worth and self-acceptance in discussion could be better articulated, it seems to be extremely interesting in the frame of this study and future research directions.

For better reader navigation, I would suggest dividing the discussion and shaping separate conclusions with subsections of limitations and future research directions.

I hope you find my comments helpful.

Author Response

Response to Reviewer 1 Comments

  1. Summary

Thank you very much for taking the time to review this manuscript. The authors have carefully read and addressed all your comments. Please find the detailed responses below and the corresponding revisions/corrections highlighted/in track changes in the re-submitted file.

  1. Point-by-point response to Comments and Suggestions for Authors

Comments 1: I am delighted to review an original article draft entitled "Direct and indirect effects of youth sports participation on emotional intelligence, self-esteem and life satisfaction" focused on the mediating effects of emotional intelligence and self- 13 esteem between youth sports participation and their life satisfaction, as well as the comparative 14 effects of different types of sports involvement. In general, the paper is logically structured and well-written. The abstract is informative. However, there is room for improvement.

Response 1: Thank you very       much for your comments that helped us improve the manuscript.

Comments 2: I have spotted a few redundant capital letters, for example, in line 56.

Response 2: Thank you for pointing out this issue. These capital letters are related to the acronym HBSC that was included in the revised version of the manuscript (please see line 64).

Comments 3: Due to complexity, the variable "satisfaction" could be better introduced in the introduction, referring to recent studies and conceptual works.

Response 3: The authors thank you for highlighting this issue. The paragraph from line 53 to 62 was revised to better introduce the concept of life satisfaction and its differentiation from the affective dimension. More studies were also included, presenting findings of the correlates and outcomes of life satisfaction in youths.

Comments 4: (67-107) in general, the proposed theoretical framework could be enhanced by deeper diving into self-concept and self-esteem, e.g. please see Glebova & Lopez Carril, 2023 and Hidayat & Yudiana & Hambali & Sultoni et al., 2023

Response 4: As suggested by the reviewer, the authors have provided further information on self-esteem concept and theoretical framework, as well as its effects on well-being and mental health (please see lines 91-95). Unfortunately, we were not able to properly identify and include the suggested references. However, two other references of special interest were included.

Comments 5: Self-worth and self-acceptance in discussion could be better articulated, it seems to be extremely interesting in the frame of this study and future research directions.

Response 5: Thank you for pointing out this issue. The authors agree with this comment and, therefore, have included the specific indirect effect of self-esteem, as well as the articulation between the emotional intelligence and life satisfaction variables (please see lines 413-417).

Comments 6: For better reader navigation, I would suggest dividing the discussion and shaping separate conclusions with subsections of limitations and future research directions. I hope you find my comments helpful.

Response 6: In line with your comment, we have divided the Discussion in subsections and included a new section for the Conclusions.

Reviewer 2 Report

Comments and Suggestions for Authors

I enjoyed reading your manuscript. My hope is my comments are viewed as helpful.

Abstract, please provide some basic statistics in your results sentences. For instance, you could place in the total indirect effect statistics.

Keywords, I believe the point of keywords is to use words not in the title and hence in search engines, the search is enhanced. You could use psychological health, competitive athletics, direct and indirect effects as potential examples.

Introduction, you could/should update your information by using the Portugal Global Matrix 4.0 information - https://www.activehealthykids.org/portugal/

Reasonable to look at the following new meta-analysis as it includes satisfaction with life/sport domain type of outcomes to update your introduction.

Lochbaum, M., & Sisneros, C. (2024). A systematic review with a meta-analysis of the motivational climate and hedonic well-being constructs: the importance of the athlete level. European journal of investigation in health, psychology and education, 14(4), 976-1001.

The importance of well-being in sport is in this manuscript. Again, it is reasonable to update your introduction with it.

Front. Sports Act. Living, https://doi.org/10.3389/fspor.2023.1256490 Clarifying concepts: “Well-being” in sport

Methods, that is a sufficient sample size to help with all your analyses.

2.1., I think this could all be one paragraph.

I think this sentence should be in your procedures section - This study conformed to the Declaration of Helsinki and was approved by the ethics committee of the Portuguese Directorate-General for Education (Study Registration N0395700001/MIME).

2.3., one paragraph with the addition of the above sentence, seems like a good idea.

2.4., I think you should describe your inspection statistics to go along with your sentence. A preliminary inspection of the dataset was performed to screen the data for accuracy, normality of the distributions and missing values.

I guess it goes within the next paragraph.

Table 1, please provide the range of scores for each variable and 95% confidence intervals.

It seems you can merge this sentence into the next paragraph.

Table 3 summarizes the descriptive statistics for the dependent variables by type of sports participation.

Separate one-way ANOVAs…

Discussion

Please expand your last sentence with details practical suggestions for coaches.

Sports coaches and researchers are encouraged to implement emotional education programmes within youth sport contexts to foster and promote emotional and psychological well-being during adolescence.

Comments on the Quality of English Language

Minor edits required.

Author Response

Response to Reviewer 2 Comments

  1. Summary

Thank you very much for taking the time to review this manuscript. We have carefully read and addressed all your comments. Please find the detailed responses below and the corresponding revisions/corrections highlighted/in track changes in the re-submitted file.

  1. Point-by-point response to Comments and Suggestions for Authors

Comments 1: I enjoyed reading your manuscript. My hope is my comments are viewed as helpful.

Response 1: Thank you very       much for your comments that helped us to improve the manuscript.

Comments 2: Abstract, please provide some basic statistics in your results sentences. For instance, you could place in the total indirect effect statistics.

Response 2: Thank you for pointing this out. However, due to the complexity of the model tested and several results, we think it would be difficult to select and report only some basic statistics. Therefore, the authors decided to maintain the Abstract as presented at the original version.

Comments 3: Keywords, I believe the point of keywords is to use words not in the title and hence in search engines, the search is enhanced. You could use psychological health, competitive athletics, direct and indirect effects as potential examples.

Response 3: As suggested by the reviewer, new keywords were included (please see line 27).

Comments 4: Introduction, you could/should update your information by using the Portugal Global Matrix 4.0 information - https://www.activehealthykids.org/portugal/

Response 4: Thank you for the suggestion. The reference was included in the study (please see line 38).

Comments 5: Reasonable to look at the following new meta-analysis as it includes satisfaction with life/sport domain type of outcomes to update your introduction.

Lochbaum, M., & Sisneros, C. (2024). A systematic review with a meta-analysis of the motivational climate and hedonic well-being constructs: the importance of the athlete level. European journal of investigation in health, psychology and education, 14(4), 976-1001.

Response 5: While we appreciate the reviewer’s comment, we respectfully disagree. Although the suggested meta-analysis reports on life satisfaction, it does not include the other variables that are central to our study (i.e., emotional intelligence and self-esteem). Therefore, we chose to not include and refer this meta-analysis.

Comments 6: The importance of well-being in sport is in this manuscript. Again, it is reasonable to update your introduction with it.

Front. Sports Act. Living, https://doi.org/10.3389/fspor.2023.1256490 Clarifying concepts: “Well-being” in sport

Response 6: Thank you for pointing this out. The recommended study aims to clarify the use and measurement of different well-being perspectives in the field of sport and exercise. Our introduction clearly states that the scope is subjective well-being (hedonic perspective), more precisely life satisfaction. Therefore, we think that our introduction introduces sufficiently the concept of well-being. Moreover, and according to one of the reviewer’s 1 comments, we have included new information on this issue (please see lines 54 to 62).

Comments 7: Methods, that is a sufficient sample size to help with all your analyses.

Response 7: Thank you!

Comments 8: 2.1., I think this could all be one paragraph.

Response 8: Thank you for the suggestion. The subsection 2.1 was merged in one paragraph.

Comments 9: I think this sentence should be in your procedures section - This study conformed to the Declaration of Helsinki and was approved by the ethics committee of the Portuguese Directorate-General for Education (Study Registration N0395700001/MIME).

Response 9: Thank you for the suggestion. We have included this sentence in the procedures section (please see lines 209 to 211).

Comments 10: 2.3., one paragraph with the addition of the above sentence, seems like a good idea.

Response 10: Thank you for the suggestion. However, in this case, the authors have maintained the subsection 2.3 in different paragraphs.

Comments 11: 2.4., I think you should describe your inspection statistics to go along with your sentence. A preliminary inspection of the dataset was performed to screen the data for accuracy, normality of the distributions and missing values. I guess it goes within the next paragraph.

Response 11: Thank you for pointing this out. For clarity purposes, we maintained the sentence but removed the information regarding the normality assessment.

Comments 12: Table 1, please provide the range of scores for each variable and 95% confidence intervals.

Response 12: The range of scores and 95% confidence intervals were computed and reported in Table 1 as requested.

Comments 13: It seems you can merge this sentence into the next paragraph.

Table 3 summarizes the descriptive statistics for the dependent variables by type of sports participation. Separate one-way ANOVAs…

Response 14: Thank you for pointing this out. We have changed the position of this paragraph, so it may appear after Table 3.

Comments 14: Discussion. Please expand your last sentence with details practical suggestions for coaches.

Sports coaches and researchers are encouraged to implement emotional education programmes within youth sport contexts to foster and promote emotional and psychological well-being during adolescence.

Response 14: Thank you for the suggestion. We have included some practical suggestions for coaches, as requested (please see lines 467 to 470).

Reviewer 3 Report

Comments and Suggestions for Authors

Thank you for the opportunity to rate this article. The authors have done a good research and the article leaves a good impression. Its main strengths are: clear research idea, concentrated justification of the research problem, clear hypothetical model, clearly described data collection instruments. Although the measures used have been validated in previous studies, the authors of the article also checked the psychometric characteristics of measures. The results are presented clearly and the conclusions are scientifically sound.

I would like to offer some recommendations for the authors of the article.

The statements in lines 45-48 could be based on specific sources (please insert references if possible).

The word "random" is not very specific (what kind of random sampling was used?) lines 187-191.

The study sample is quite impressive. Information about nonparticipants, sport participants was provided. Also information about sport type is provided. But no information about sport experience. If You have such information, please provide it. 

Author Response

Response to Reviewer 3 Comments

  1. Summary

Thank you very much for taking the time to review this manuscript. We have carefully read and addressed all your comments. Please find the detailed responses below and the corresponding revisions/corrections highlighted/in track changes in the re-submitted file.

  1. Point-by-point response to Comments and Suggestions for Authors

Comments 1: Thank you for the opportunity to rate this article. The authors have done a good research and the article leaves a good impression. Its main strengths are: clear research idea, concentrated justification of the research problem, clear hypothetical model, clearly described data collection instruments. Although the measures used have been validated in previous studies, the authors of the article also checked the psychometric characteristics of measures. The results are presented clearly and the conclusions are scientifically sound. I would like to offer some recommendations for the authors of the article.

Response 1: Thank you very       much for your comments that helped us improve the manuscript.

Comments 2: The statements in lines 45-48 could be based on specific sources (please insert references if possible).

Response 2: Two specific sources were included, as suggested (please see line 52).

Comments 3: The word "random" is not very specific (what kind of random sampling was used?) lines 187-191.

Response 3: We agree with this comment, and we have presented more specific information (please see line 203).

Comments 4: The study sample is quite impressive. Information about nonparticipants, sport participants was provided. Also information about sport type is provided. But no information about sport experience. If You have such information, please provide it.

Response 4: Unfortunately, that information was not collected.

Round 2

Reviewer 1 Report

Comments and Suggestions for Authors

Dear Authors,

Thank you for the effective revisions.

Reviewer 2 Report

Comments and Suggestions for Authors

Thank you for your revised manuscript. I believe you addressed my comments in full. Thank you for doing so.

Comments on the Quality of English Language

Minor edits.

Reviewer 3 Report

Comments and Suggestions for Authors

The authors of the article take into account the comments, the article has been improved